# Different Polymers for the Base of Removable Dentures? Part II: A Narrative Review of the Dynamics of Microbial Plaque Formation on Dentures

**DOI:** 10.3390/polym16010040

**Published:** 2023-12-21

**Authors:** Pierre Le Bars, Alain Ayepa Kouadio, Yves Amouriq, François Bodic, Pauline Blery, Octave Nadile Bandiaky

**Affiliations:** 1Department of Prosthetic Dentistry, Faculty of Dentistry, Nantes University, 1 Place Alexis Ricordeau, F-44042 Nantes, France; ayepa_alain@yahoo.fr (A.A.K.); yves.amouriq@univ-nantes.fr (Y.A.); francois.bodic@univ-nantes.fr (F.B.); pauline.blery@univ-nantes.fr (P.B.); 2Nantes University, Oniris, University of Angers, CHU Nantes (Clinical Investigation Unit Odontology), INSERM, Regenerative Medicine and Skeleton, RMeS, UMR 1229, F-44000 Nantes, France; octave.bandiaky@univ-nantes.fr; 3Department of Prosthetic Dentistry, Faculty of Dentistry, CHU, Abidjan P.O. Box 612, Côte d’Ivoire

**Keywords:** *Candida* spp., dental plaque biofilm, denture management, denture hygiene, denture stomatitis, microbiome, systemic

## Abstract

This review focuses on the current disparities and gaps in research on the characteristics of the oral ecosystem of denture wearers, making a unique contribution to the literature on this topic. We aimed to synthesize the literature on the state of current knowledge concerning the biological behavior of the different polymers used in prosthetics. Whichever polymer is used in the composition of the prosthetic base (poly methyl methacrylate acrylic (PMMA), polyamide (PA), or polyether ether ketone (PEEK)), the simple presence of a removable prosthesis in the oral cavity can disturb the balance of the oral microbiota. This phenomenon is aggravated by poor oral hygiene, resulting in an increased microbial load coupled with the reduced salivation that is associated with older patients. In 15–70% of patients, this imbalance leads to the appearance of inflammation under the prosthesis (denture stomatitis, DS). DS is dependent on the equilibrium—as well as on the reciprocal, fragile, and constantly dynamic conditions—between the host and the microbiome in the oral cavity. Several local and general parameters contribute to this balance. Locally, the formation of microbial plaque on dentures (DMP) depends on the phenomena of adhesion, aggregation, and accumulation of microorganisms. To limit DMP, apart from oral and lifestyle hygiene, the prosthesis must be polished and regularly immersed in a disinfectant bath. It can also be covered with an insulating coating. In the long term, relining and maintenance of the prosthesis must also be established to control microbial proliferation. On the other hand, several general conditions specific to the host (aging; heredity; allergies; diseases such as diabetes mellitus or cardiovascular, respiratory, or digestive diseases; and immunodeficiencies) can make the management of DS difficult. Thus, the second part of this review addresses the complexity of the management of DMP depending on the polymer used. The methodology followed in this review comprised the formulation of a search strategy, definition of the inclusion and exclusion criteria, and selection of studies for analysis. The PubMed database was searched independently for pertinent studies. A total of 213 titles were retrieved from the electronic databases, and after applying the exclusion criteria, we selected 84 articles on the possible microbial interactions between the prosthesis and the oral environment, with a particular emphasis on *Candida albicans*.

## 1. Introduction

The dynamic and fragile balance of the oral cavity ecosystem depends on pH, thermal fluctuations, humidity, enzymes, and microflora [1]. Against the backdrop of these conditions, there is an interaction between the oral environment and the physical and chemical characteristics of the basic materials of prostheses [2]. The traditional polymerization reaction of polymer chains proceeds with increasing heat until the monomers transform into a polymer. However, the use of this technique produces residual monomers that can negatively affect the physical, mechanical, and biological properties of the base of the prosthesis [3,4]. To remedy this, new hardening procedures have recently emerged. Therefore, the use of processing techniques such as injection molding, microwave energy, autoclaving, high-pressure thermal polymerization, CAD/CAM milling, and 3D printing have been proposed [5,6].

A thorough long-term in vivo evaluation to verify that the different basic thermoplastic resins for removable prostheses are biocompatible and exhibit insignificant cytotoxicity remains to be carried out. In fact, these base resins for prostheses that are in permanent contact with the mucous membranes can release cytotoxic components locally, causing irritation and inflammation [7].

Immediately after brushing or prophylaxis, the denture in the mouth is covered with a salivary pellicle, which precedes colonization by the first pioneer bacteria. Subsequently, the succession of early (*Streptococcus* species) and late colonizers in the biofilm, under optimal conditions, will favor the survival of new species. Microorganisms from the biofilm on the denture surface can penetrate the different polymer biomaterials.

A recent in vitro study tested the polymethyl methacrylate (PMMA) denture base material Vertex RS (Vertex-Dental, Soesterberg, The Netherlands) immersed for 30, 60, and 90 days in a suspension of *Candida albicans*. The authors highlighted blastospores and pseudohyphae on the surface of this material, which were detected in the crystallized structures as well as in traces after grinding. These authors put forward the hypothesis that the penetration of *C. albicans* stems from the deterioration of the material surface, leading to the formation of microporosities, which makes disinfection difficult and thus facilitates recolonization [8].

In vivo analysis of the interactions between the denture surfaces, saliva, eukaryotic and prokaryotic microorganisms that can cause infections such as denture stomatitis (DS) is of great importance for the prevention and treatment of these pathologies. Among the eukaryotes, *Candida* species have been reported to have the ability to attach to bacterial biofilms at almost every stage of formation, referred to as a “mycofilm”.

The behavior of this mycofilm is modified and fluctuates depending on the properties of the surface of the polymers, the interactions between the microorganisms, the architecture of the biofilm, and the saliva and environmental conditions, with the last two being dependent on the general state of the prosthesis wearer [5].

This review aims to shed light on the indications for the different polymers used in the composition of prosthetic bases from a biological point of view considering the oral cavity. The behavior of these materials with respect to microorganisms depends on the adhesion, aggregation, and accumulation of prosthetic microbial plaque. These phenomena depend on the state of the surface of the materials (roughness, wettability, and free energy) but also on the patient’s hygiene with respect to the prosthesis, on the clinical need for relining, and on the general state of the patient. The objective of this review is to provide an update on the specificities of polymers (PMMA, polyamide (PA), and polyether ketone (PEEK)) used as a prosthetic base to facilitate the maintenance of a healthy oral environment.

## 2. Materials and Methods

The methodology used for this review comprised the formulation of a search strategy, with inclusion and exclusion criteria defined and applied to retrieve studies. After selecting relevant studies, data were extracted to summarize the results. The PubMed database was searched to gather the relevant literature published on the topic. The search terms used were “dental plaque biofilm”, “*Candida* spp.”, “denture management”, “denture hygiene”, and “denture stomatitis.” The inclusion criteria were (a) articles written in English that (b) dealt with the microbial flora interactions between the polymer in the denture base and the oral environment, with the occurrence of DS, and (c) articles that reported on the control of the microbial plaque of dentures (DMP). Articles that did not meet the predetermined inclusion criteria were excluded and the articles selected for the final analysis were obtained as full text.

## 3. Results

The current analysis focused on prosthetic microbial plaque and considered different parameters that can influence this type of colonization. Given the abundance of data obtained, we grouped the results by theme into five tables and two Appendix A. Table 1 describes the chemical composition, roughness (Ra), and surface free energy (SFE) of the polymers (PMMA, PA, and PEEK). These parameters have the potential to affect bacterial adherence [9]. Indeed, a low value of SFE is sought to resist plaque in in vitro studies. Concerning the critical surface energy of acrylic materials, the zone of good adhesion is located at values greater than 40 mJ/m^2^ [10]. Table 2 lists the different protocols for polishing the PMMA, PA, and PEEK polymers of the prosthetic base (mechanical polishing and/or chemical polishing). CAD/CAM-milled acrylic resins have lower Ra values than heat-cured PMMA. Mechanical polishing of PMMA is superior to chemical polishing. Polishing the PAs makes it possible to obtain a roughness close to 0.2 µm. Chairside polishing of PEEK also makes it possible to obtain clinically acceptable values. Table 3 describes the frequency and protocol for cleaning dentures. A protocol of daily cleaning for dentures is recommended for the three polymers. The cleanser tablets tested were more effective for PMMA and PEEK than for thermoplastic polyamide. On the other hand, self-polymerized and injection-molded polyamide showed higher solubility than PMMA. Concerning PEEK, the cleaning tablets were effective wit, low solubility. Generally speaking, denture cleansers increased the roughness of all PMMA. Concerning liquid cleansers, the best result was obtained with 2% CHG and 0.5%–1% NaOCL for PMMA. Thermo-injected polyamide base resins for prostheses colonized by *C. albicans* and disinfected with 0.12% chlorhexidine and Neem demonstrated the highest antimicrobial level. Table 4 presents the data on denture relining. Good relining was obtained with conventional thermoset PMMA and the CAD/CAM-milling block. PAs had low adhesion strength, PEEK required specific preparation and exhibited a mixed type of failure involving adhesion and cohesion. Table 5 shows a synthesis of the results concerning polymers (PMMA, PA, PEEK) for denture microbial plaque formation, polishing, relining, and hygiene. Overall, PMMA presented advantages over PA and PEEK in most of the sections mentioned. In vitro coating or the addition of antimicrobial components was desirable with PMMA. The effects of the incorporation of different nanoparticles (AgNP, silver-zinc zeolite, TiO_2_ and Fe_2_O_3_) or natural compounds such as oleic acid (OA), in PMMA produces antifungal and antibacterial effects, limiting the development of biofilm on the surface of the prosthesis. However, if the dosage of these components was not respected, harmful mechanical effects were observed, as well as undesirable cytotoxic effects (Appendix A). The in vitro comparison between PMMA and polyamide regarding cytotoxicity did not reveal obvious differences. Studies remain disparate with regard to the materials studied and the protocols used. The results fluctuated depending on the duration of the experiments and different parameters such as temperature and surface condition (Appendix A).

## 4. Discussion

A removable prosthesis residing in the oral cavity exposes the existing planktonic microbiota (bacteria, archaea, viruses, and eukaryotic organisms) to stress [78,79,80]. These conditions are favorable for the growth of DMP [81,82,83]. Quantitatively, this biofilm is defined as a community of more than 10^11^ microorganisms per gram of dry weight [84,85], attached to the extrados and intrados of the surface of the prosthesis and sur-rounded by an extracellular matrix (ECM) produced by the bacteria and *Candida* themselves [86,87]. This matrix, composed of macromolecules such as exopolysaccharides, proteins, and DNA [88], provides structural integrity to the biofilm and offers a physical barrier that may be impenetrable to drugs.

In contact between the soft tissues, living tissue and the inert polymer provide another favorable environment in the oral cavity for microbial colonization [89,90,91]. At the level of the intrados, this decreased space leads to a reduction in oxygenation, salivary flow, and pH, which promotes the activity of secreted aspartyl proteinases (SAPs) in the matrix. This environment plays a central role in the pathogenicity of *Candida* [92,93,94].

The maturation of the *C. albicans* biofilm proceeds according to the same steps but more slowly than the bacterial biofilm. The presence of hyphae and pseudohyphae is the main difference between the two biofilms. Recent targeted studies have explained the initial adhesion to the prosthetic surface, the subsequent development of mature biofilms [95], the formation of the extracellular matrix, and finally, the dispersal mechanism [96,97,98] (Figure 1).

Up to three quarters of patients who wear removable prostheses can develop an inflammation called “denture stomatitis” (DS). This pathology is characterized by an imbalance of the microbial flora or dysbiosis, resulting simultaneously in an abundance of opportunistic pathogens such as *C. albicans* [99,100], the differential proliferation of certain bacterial species determined using culture and next-generation sequencing (NGS) [101,102,103,104,105,106,107,108,109], and a decrease in microbial diversity [102,103,104].

Dental surgeons aware of the risk posed by this infectious condition to vulnerable patients should regularly check the oral health of users of removable prostheses [110]. For this, although DMP cannot be totally eradicated, it can be controlled through oral hygiene practices that include a daily regimen of brushing the mucous membrane and the denture, followed by rinsing with an antiseptic mouthwash [111,112,113]. Maintaining a healthy state helps to avoid the transition from a harmless commensal to a pathogen.

An oral hygiene regime adapted to the different polymers requires knowledge of the particularities of the materials used as well as the effects of their modifications (polishing and relining) on the oral microbiota [114]. The objective of this review is to provide an update of the specificities of the polymers (PMMA, PA, and PEEK) used in prosthetic bases to help facilitate the maintenance of a healthy oral environment.

Several current precautions and methods make it possible to limit the drift of the oral microbiota toward dysbiosis in wearers of removable prostheses.

### 4.1. Polymers in the Oral Environment

Once it is introduced in the mouth, a denture is rapidly coated by saliva and constitutes the ideal platform for dynamic microbial growth of DMP [115,116,117]. These biofilms represent a wide range of microorganisms, comprising all three domains of life. Their proximity to the denture polymer offers numerous possibilities for physical and chemical interactions between different species and kingdoms (Delaney, C.; 2019) [118]. On the other hand, the interaction between the prosthetic base and the biofilm on the surface of the oral mucosa can favor the release of potentially toxic substances from the polymer that in turn interact with the host tissues [119].

Biofilm development under an acrylic denture increases the risk of DS fivefold com-pared with a metallic denture [120]. Another drawback associated with poor denture hygiene is bad breath, which can be the cause of patient discomfort [121]. These bad odors are related to the microbial plaque of the denture [122]. Studies using new technologies (next-generation sequencing, NGS) in the field of bacterial identification highlighted the emergence of the phyla Firmicutes and Fusobacteria and the genera Leptotrichia, Atpobium, Megasphaera, Oribacterium, and Campylobacter as being associated with the bad smell of prostheses. Here, good oral hygiene is essential to combat bad odors [123].

In DS, lack of or ineffective brushing in the absence of a cleanser promotes the rapid growth of biofilm on the surface of prostheses [9,10]. Clinically, the selection of polymer used for the prosthetic base must consider the adhesion of microorganisms. This colonization promotes the penetration of the microbiota and reduces the fracture resistance of prostheses [124].

#### 4.1.1. Polymer and Microbial Adhesion

A roughness (Ra) promotes adhesion and bacterio-fungal aggregation on acrylic resins [125]. However, some authors point out that the initial colonization does not differ in accordance with the range of dental materials [126,127]. In the same way, research has not highlighted a link between the roughness of the surface, the hydrophobicity/hydrophilicity of the acrylic resin, and the metabolic activity of adherent *C. albicans* cells [128]. Aggregation of *C. albicans* with other microorganisms and the influence of saliva, through its antimicrobial power, flow, and composition, seem to dominate the conditions of adhesion to the surface of a prosthesis (roughness and SFE) [129]. For other authors, *Candida* adhesion was strongly affected by Ra, saliva, and bacteria, but not by SFE [125]. Despite this discrepancy, the results suggest that a reduction in the *C. albicans* biofilm may be related to modifications of the surface of the PMMA thanks to the coating. The coating promotes hydrophilicity and in addition to the influence of roughness [130]. In addition, the DMP is subject to various mechanical constraints such as food tenacity, temperature fluctuations, chewing forces, and the load of the prosthetic device [104,105,106,107,108,109,110,111,112,113,114,115,116,117,118,119,120,121,122,123,124,125,126,127,128,129,130,131]. Microbial adhesion has been studied in relation to PMMA, in particular, and much less so in relation to PA and PEEK.

#### 4.1.2. PMMA and Adhesion

PMMA is naturally hydrophobic [19] Gad MM, 2022, but this material, which is used in the composition of dentures, contains many carboxylate and methyl ester groups. This chemical composition, on the one hand, accounts for the hydrophilic nature of the dentures and, on the other hand, produces a large amount of SFE. In vitro, the adhesion of Pseudomonas fluorescens proved to be favorable to hydrophobic surfaces, with the lowest adhesion threshold for a roughness of 0.4 μm. Although the weakest adhesion of mammalian cells occurred at a roughness of 0.1 μm, the latter was favored in the presence of hydrophilic surfaces (PMMA) Choi SY, 2016 [132]. However, the variations in the chemical composition of the material used for the denture base partly explain the disparity in characteristics between the different brands of PMMA on the market Sipahi, 2001 [133]. Compared to the traditional fabrication method, acrylic resin injection offers a reduction in the surface roughness of the prosthesis base as well as decreased bacterial adhesion [21] Moslehifard E, 2022 (Table 1).

#### 4.1.3. Polyamide and Adhesion

Analysis of the adhesion of microorganisms, in particular, yeasts, to PA remains very limited. Nevertheless, an experiment conducted on the effect of a prosthetic cleanser on the formation of a mycofilm on a PA resin (Flexite MP) and a polymethyl methacrylic resin (Acron MC) showed that *C. albicans* had a significantly higher growth rate on PA than on PMMA de Freitas Fernandes FS [134].

As a crystalline polymer, PA has better biocompatibility for patients who are allergic to acrylic resins. But over time, PA has significant disadvantages, displaying high water absorption, increased solubility, an overly rough surface, and bacterial contamination. In addition, this material remains difficult to polish and may result in color deterioration in the mouth Vojdani, 2015 [135]. Higher microbial adhesion was recently observed on injection-molded PA than PMMA (Table 1) [16] Sultana, 2023.

In order to remedy this, minimal changes in the injection manufacturing protocol of two PA prosthetic base materials were tested in vitro (Perflex Biosens (BS), Netanya, Israel and VertexTM ThermoSens (TS), Soesterberg, The Netherlands. By slightly modifying the melting temperature (5 °C) and pressure (0.5 bar), no improvement in the surface finish was observed for Biosens, whereas for ThermoSens, the surface roughness was significantly reduced Chuchulska, 2022 [136].

#### 4.1.4. PEEK and Adhesion

As early as 2007, Kurtz et al. emphasized the non-allergenic properties of PEEK and its low affinity for dental plaque. PEEK is considered hydrophobic and has a low SFE. As a result, *C. albicans* adhesion is facilitated [27,137]. This was compared to the formation of biofilm on the surface of different materials in vitro (zirconia, titanium, PMMA, and PEEK). In their study, PEEK and PMMA yielded the same results but were linked to less biofilm formation than zirconia and titanium. However, the surface condition of PEEK was smoother than that of zirconia and titanium [138]. It has been reported that PEEK has good biocompatibility in vitro and in vivo, causing neither toxic nor mutagenic effects nor clinically significant inflammation. In addition, PEEK lends itself to sufficiently effective polishing so as to delay the fixing of microbial plaque [139]. PEEK without any additives is biologically inert and naturally hydrophobic when in contact with saliva. The 80°–90° contact angle of saliva can be reduced by adding plasma coatings, which are effective methods for modifying surface properties [140] to improve the hydrophilicity [138].

When comparing PEEK with other computer-aided design/computer-aided manufacturing (CAD/CAM) materials, PEEK samples are slightly rougher than PMMA samples. The reason is linked to the ceramic particles that are added to PEEK [25] (Table 1).

#### 4.1.5. Polymer and Accumulation of DMP

After the adhesion of the first colonizers on the denture surface, to preventively limit the accumulation of microorganisms, and particularly of *Candida* and bacteria populations, several parameters can be modified to facilitate the optimization of the manufacture of polymers. The incorporation of antifungal agents into denture base resin may reduce the colonization of *C. albicans* [141]. There are few data on PAs and PEEKs, whereas PMMAs, in contrast, have been the subject of numerous experiments (Table 3).

For example, nanoparticles (such as fluoridated apatite-coated titanium dioxide, FAp-TiO_2_) in PMMA facilitate the production of reactive oxygen species by promoting the photocatalytic effect after irradiation, which neutralizes the attachment of *C. albicans.* This effect is sought to facilitate the maintenance of removable prostheses in geriatric patients [142]. The incorporation of bioactive glass (BAG) in thermopolymerized or polymerized acrylic resins at room temperature significantly lowers the adhesion of *C. albicans*. For both types of polymerization, the hardness of acrylic resins is improved by adding BAG [140].

Another parameter can be modified to promote hydrophilicity to limit the adhesion of *C. albicans* on an acrylic resin denture with photopolymerized coating [143]: Plasma treatment of PMMA on the surface increases SFE, facilitates wettability, and lowers the contact angle, all of which reduce the adhesion of *C. albicans* [144,145]. In contrast, trimethylsilane coating increases hydrophobicity, reduces wettability of the denture base surface, and inhibits the adhesion of *C. albicans* [146]. The TiO2 coating creates a super-hydrophilic surface. It thus promotes wettability, which is essential for reducing *Candida* adhesion. The implementation of the PMMA surface coating involves only moderate costs while preserving the properties of the original material [147,148]. Recently, to assess the effectiveness and the antibacterial properties of a silver nanoparticle (NAg), a solution of NAg mixed with acrylic acid and methyl methacrylate (MMA) monomer was tested (in vitro and in vivo on animals) and compared with a PMMA solution without NAg. The results concerning the state of the prosthetic surface, the mechanical properties, the antimicrobial effect of NAg, the longevity, and the biological and toxic harmlessness of the NAg/PMMA prosthesis base were superior to the PMMA base without NAg. However, clinical confirmation must be provided by studies with humans [41,42,43,44,140,149,150,151].

#### 4.1.6. Polishing to Limit Microbial Adhesion

The adhesion of early microbial colonizers is closely related to the finish of the denture surface. This adhesion during the initial phase of microbial colonization on flexible prostheses is similar to that of acrylic resin prostheses. This result was confirmed by a laboratory study showing that acrylic resin and PA resin are easily colonized by *Candida* species. However, the growth rate of this fungus is significantly higher on PA resin than on PMMA (*p* < 0.001) [152].

Different tests of the surface condition of the material (polishing) have shown that the polishing method alone (wood sandpaper: grit 180) is essential in terms of roughness compared with the drying method of self-curing acrylic resin. Moreover, chemical polishing (at 70 °C for 10 s) aggravates the roughness [74,153]. Regarding PMMA resin, the residual monomer acts on the SFE by reducing adhesion and *Candida* growth [134]. For PA resin (Breflex polyamide, Bredent, GmbH Co. KG, Senden, Germany) fabricated using the injection-molding technique, no significant correlation was observed in contact angles for mechanical polishing versus chemical polishing. This difference was related to the specific physical properties of the materials used [31].

The design and manufacture of CAD/CAM prostheses machined from blocks of polymerized PMMA under high temperature and high pressure led to a smoother surface finish than PMMA-HC based on CAD/CAM prostheses [154]. As a result, for patients at risk of *Candida* fungal infection, the surface properties of CAD/CAM PMMA represent a possibility of reduced adhesion of this fungus (Table 2).

Quezada (2022) [37] and Corsalini (2009) [33], using the same in vitro mechanized and manual polishing methods, attempted to standardize a polishing protocol. However, since contradictory results were reported, with one favoring the manual method and the other the mechanized method, new investigations have to be carried out.

An explanation for the contradictory results is offered by previous research. The structure of PMMA directly after polymerization had a low initial roughness, and subsequent polishing made it easy to reach clinically acceptable values. On the other hand, PAs were more difficult to polish due to their fibrous semi-flexible structure and low surface hardness [155]. Although PEEK and PMMA have similar values of Vickers hardness, the composition and the state of the surface roughness differed between the two materials [156]. Therefore, surface polishing that is specific to the two materials is required.

Thus, regarding the polishing of PEEK, Kurahashi et al. (2020) [39] suggest the use of a soft brush coupled with a cleaning agent for more than 3 min to achieve clinically acceptable surface roughness (Table 2).

Heimer et al. [41] compared the effects of laboratory and chairside polishing methods on the surface roughness of PEEK and reported that chairside polishing of PEEK yielded lower surface/laboratory roughness values (Table 2).

Fused deposition modeling of PEEK is one of the most practical additive techniques; compared to other polymers, PEEK remains stable over the long term regarding its wear and color [157,158]. The biocompatibility and biostability of PEEK are supported by the U.S. FDA drug and device master files [159]. Another way to limit the initial adhesion of microorganisms and particularly of *C. albicans* on the prosthetic surface is to use a coating.

#### 4.1.7. Denture Base Surface Coating to Limit Adhesion

Among the types of coatings available, cold plasma under heat-polymerized acrylic resin prevents the early adherence of *C. albicans* [160]. Another goal for coating the polymer (PMMA) with creamers is to enhance the resistance of the denture base surface. Indeed, coating creamers (inorganic–organic hybrid polymeric) enhance the scratch resistance of PMMA denture resin (increasing the flexural strength (FS), flexural modulus (FM), and hardness) [161,162]. To date, in view of the diverse results of experiments, no consensus has been reached on this topic. To fight against the adhesion of *Candida*, the surface of the denture base must be smooth, hydrophilic, and without roughness. Further investigations are needed to better understand the correlation between factors affecting the hydrophobicity of the denture base and the adhesion of *C. albicans.*

#### 4.1.8. Effects of Cleaning on Denture Materials (Table 3)

Currently, the use of a prosthesis cleanliness index makes it possible to assess the hygiene of prostheses by visualizing the quantity of stains on the intrados of the denture. Rinsing beforehand eliminates invisible microbial plaque. The scores, ranging from 0 (best) to 4 (worst), help to adapt the hygiene instructions for the wearers of dental prostheses [163].

The use of bleach-based cleansers, according to the recommended dosages (containing 1.5% or 2% *w*/*v* sodium hypochlorite and/or 1.7% *w*/*v* sodium hydroxide) and duration of use (at least 3 min daily), is associated with sufficient antimicrobial activity against *Streptococcus mutans* and *C. albicans*, without any changes to acrylic color, surface roughness, or mechanical properties [164,165]. However, in the long term, these cleansers corrode and tarnish metal prostheses. Effervescent cleansers have also proven their effectiveness, but they are not recommended in the presence of prosthesis relining materials.

Manual brushing with a toothbrush plus soap and water is the most common method for maintaining removable dentures (Milward P, 2013) [166]. Several adjuvants to increase the effectiveness of manual cleaning in the form of pastes, gels, foams, and powders are on the market [167].

The use of antiseptics to inhibit or eliminate microorganisms and immersion in a chemical solution for 8 h are recommended. Sodium hypochlorite, chlorhexidine diglconate, and alcohol can disinfect or reduce the dental plaque on acrylic resin dentures without being cytotoxic [110,112,113]. The different methods of cleaning dentures can influence the physical and aesthetic characteristics of the prosthesis materials. Also, in order to ensure the clinical durability of removable prostheses, patients and clinicians should be aware of the manufacturer’s instructions for use [168].

Although there is no consensus regarding how to best maintain prosthetic hygiene compatible with the patient’s state of health [61], the disadvantages of many procedures have been thoroughly evidenced [169].

Hydrogen peroxide-based disinfectants should not be used regularly, as they cause surface roughness of the PMMA. NaOCl is less aggressive and generates slight alterations on the surface of the prosthetic base [170]. In addition, sodium hypochlorite was found to be non-cytotoxic after six months of use [171].

Flexural strength is reduced by immersion cleaning of removable PMMA prostheses modified with nano-ZrO_2_. Thus, a significant decrease in this resistance after immersion in different denture cleansers was reported, which was strong for sodium hypochlorite, intermediate for Corega, and low for Renew [170,172,173]. Several habits should be avoided, such as rinsing with boiling water and prolonged maintenance in a dry atmosphere or water, because these alter the qualities of PMMA and promote microbial colonization. Both bleach and isopropyl alcohol (IPA) are highly antimicrobial, but bleach is incompatible with components of metal dental prostheses and IPA mouthwashes damage PMMA [174].

Concerning denture cleaning tablets, the polarity of the resins, the concentrations of the tablets, and the chemical content of the cleanser may directly affect the formation of *C. albicans* biofilm [68]. Thus, the dosage and prescription of disinfecting tablets can vary depending on the resin used to make the prosthetic base. In tablet form, Polident^®^ has been proven to be effective as a denture cleanser. But after 30 days of immersion in a solution based on Polident^®^, the heat-polymerized acrylic resin may undergo alterations to its physical and mechanical properties. This may be related to the accelerated aging of resins caused by chemicals found in denture cleansers [175].

The mechanical properties of PEEK do not change during the sterilization process. An in vitro study showed that the solubility of PEEK in physiological saliva and distilled water is lower than that of PMMA [156]. In the study by Demirci under the same conditions, the solubility values of PEEK in distilled water were found to be similar to those of PMMA (HP: Ivoclar Vivadent AG., Schaan, Liechtenstein). In the presence of a cleanser (Corega tablet, Protefix tablet (PT), 1% sodium hypochlorite (NaOCl)), the solubility values of PEEK were found to be lower than those of PMMA. In this study, higher water sorption and solubility values were observed than those obtained by Lieberman [156]. The explanation proposed mentions the consequences of the effects of cleansers on PEEK and PMMA surfaces for 120 days. Thus, for these authors, the water sorption and solubility values of PEEK can be attributed to the molecular imbalance occurring on the surface of the PEEK [62].

The use of microwave disinfection in combination with denture cleansers and brushing has also been shown to effectively disinfect dentures, although microwaves may also physically distort denture resin [176]. The personalized implementation of the currently available means for disinfection is informed by the general condition of the patient, the material composition of the prosthetic base, and the presence or absence of DS.

### 4.2. Denture Base Relining (Table 4)

After some time (following bone resorption), it is necessary to reline the intrados in order to improve the stability, support, and retention of removable dentures. There are several commonly used relining materials, such as cold or hot polymerization, polymerization in visible light, and acrylic resins polymerized in microwaves [177,178].

At the interface between the reliner and the prosthetic base, the bond strength depends on the chemical composition of the two materials that come into contact with each other [179]. The bonding strength can be improved by treating the two surfaces that are in contact with each other [179,180,181,182]. The parameter characteristic of relining is the shear bond strength (SBS). This parameter is better for relining using thermosetting resin as well as both CAD/CAM and conventional thermosetting denture resin compared to self-curing relining resin [183,184]. An in vitro study showed that reliners with thermopolymerizable acrylic resins had an increased SBS compared to reliners with self-curing acrylic resins. This also applied to bases of conventional dental prostheses and CAD/CAM but without a significant difference. However, there was a significant difference between autopolymerizing acrylic resin bond strength with CAD/CAM and conventional denture bases.

Autopolymerizing reliner material seems to produce a stronger bond with CAD/CAM denture bases. It has been pointed out that self-curing relining material appears to produce a significantly stronger bond with a CAD/CAM denture base compared to a conventional resin base [184]. Recently, various in vitro tests of the adhesion of composite materials on thermosetting resins, on CAD/CAM, and on printed groups yielded the following results: In order of the best performance regarding the adhesion of high-viscosity/low-viscosity composites (SR Nexco, high viscosity (SR); and Kulzer Creactive, low viscosity (K)), the thermosetting resin group was first, followed by the CAD/CAM group, and finally the 3D-printed groups. However, the differences noted between these groups were not significant [185].

To assess the maintenance of rebased resin bases, five disinfectant solutions were tested: sodium hypochlorite, sodium perborate, chlorhexidine gluconate, apple vinegar, and distilled water. A prosthesis base (Vipi Wave) rebased with an acrylic resin (Tokuya-ma Rebase Fast II) after dipping showed alterations in its roughness regardless of the solution used [173,186]. Kim et al. tested relining using two hard resins, one of the self-hardening type (Tokuyama rebase II) and the other of the light-activated type (Mild Rebaron LC). They carried out these two relinings on a thermoplastic polyamide resin (Biotone; BT), on a classic thermopolymerizable acrylic resin (Paladent 20; PAL20), and, finally, on a thermoplastic acrylic resin (Acrytone; ACT). The results showed that the thermoplastic polyamide resin (Biotone) had the lowest adhesion strength of the three materials tested [69].

More recently, Vuksic Josip et al. (2023) [70] tested relining (with a soft denture liner and a silicone-based, direct relining method) on several resins: (1) Meliodent heat cure (Kulzer, Hanau, Germany), denture base material, PMMA, heat-cured; (2) Vertex Thermosens (Vetex Dental, Soesterberg, The Netherlands), denture base material, PA, technical injection; (3) three CAD/CAM subtractive materials; and (4) two CAD/CAM additive materials. With the same reliner (GC Reline II Soft), the bond strength of the PA (Vertex) and both additive manufactured denture bases was significantly lower than that of the three materials used for subtractive denture fabrication and heat-cured PMMA (Table 4). However, the authors expressed their reservations because, to date, there are only a few studies available, mainly on flexible rebasing. The tests differ between these studies, and different materials were used as controls (PMMA from different manufacturers).

The bioinert nature of PEEK can make adhesive bonding difficult. The SBS of PEEK can be increased by roughening the material or by embedding molecules on the surface through sandblasting, acid treatment, laser, or adhesive systems.

SBS values greater than 10 MPa between PEEK and resin-based composites have been reported to be clinically acceptable. However, the hydrophobic surface and low SFE of PEEK make it difficult to establish a strong and long-lasting bond. Therefore, PEEK material surface treatments and adhesive systems with resin are hot research topics focused on the application of PEEK in the restorative field. Modalities concerning the effectiveness of bonding to the surface of PEEK are not yet sufficiently developed for routine use.

### 4.3. General Conditions and Dentures

Whichever material is chosen, after adhesion, inadequate oral hygiene facilitates the accumulation of biofilm, colonizing the surface of the prosthesis. This biofilm can constitute a risk factor for infection, especially for patients who are older or who are immuno-compromised and/or have endocrine deficiency [187]. For these patients, special vigilance is necessary regarding prosthetic oral hygiene in order to avoid infectious complications.

Indeed, this additional microbial load can lead to an imbalance between bacterial species, bacteriophages, and fungi, thus promoting the resistance and virulence of mycofilms to the detriment of the host. The secretions of bacteria and fungi, by participating in the aggression of biotic surfaces (mucous membranes and teeth), promote the production of various inflammatory mediators such as cytokines [188]. The use of removable prostheses in these conditions after a certain period of time promotes bone resorption.

Some older denture wearers have medical conditions such as arthritis and dementia that can impair their ability to carry out oral hygiene procedures effectively, thus requiring assistance from caretakers and some education [189]. Specific treatments are available if a *Candida* infection is suspected [190], with accompanying denture disinfection/cleaning or replacement [8]. Other conditions such as Parkinson’s disease can lead to dentures falling out of patients’ hands because of trembling. In these cases, thanks to its flexibility, the PA prosthesis makes it possible to overcome small bone and mucous undercuts. Crossing this undercut promotes retention. The prosthetic base made of PA, due to its high resilience and impact resistance, is less prone to fractures than PMMA [191].

For patients who are hypoallergenic to prosthetic materials, different alternatives exist. Whether PMMA, PEEK, or PA, the polymerization reaction releases more or fewer toxic molecules. By dissolving in the saliva, these molecules can diffuse away from the mouth [192,193]. These are essentially, after the polymerization, the residual monomers (MMA, methyl methacrylate; BuMA, butyl methacrylate; EMA, ethyl methacrylate; EGDMA, ethylene glycol dimethacrylate) that are responsible for the toxic and allergenic effects of acrylates [194].

This residual monomer depends both on the method of polymerization (duration, cold or heat) and on the volume of the prosthetic base; it only becomes stable after 2 weeks of wearing the dentures. It is low for thermopolymerized resins and at the palatal level of the thin prosthetic base [8].

Moreover, the acidic environment and the temperature of the oral cavity promote the release of substances contained in resins such as formaldehydes, benzoyl peroxides, benzoic acid, hydroquinone, and phthalates, as well as cobalt, nickel, and beryllium. With respect to the mucous membrane, these products can cause type IV allergic reactions, or an intolerance can appear in the long term [195].

As a remedy, so-called hypoallergenic resins for dental prostheses have appeared on the market. To be suitable for hypoallergenic patients, the denture base resins should contain only a very small amount of MMA [196].

MMA can be replaced by diurethane dimethacrylate, polyurethane, polyethylene terephthalate, polyethylene terephthalate, or polybutylene terephthalate. However, only two of these have similar mechanical characteristics to PMMA resin standards: Polyan Plus^®^ and TMS Acetal Dental [195].

The in vitro comparison between PMMA and PA regarding cytotoxicity has not revealed any obvious differences. Findings remain disparate about the materials studied and the protocols used. The results vary depending on the duration of the experiments and on the different parameters analyzed, such as temperature and surface condition supplement II. For patients with low stress tolerance and sensitivity to metallic materials, PEEK is indicated for partial removable prostheses. PA bases are also an alternative for patients who are allergic to other denture base materials and for patients with microstomia [26].

## 5. Conclusions

Regarding the choice between different polymers and in view of the complexity of DMP, we still lack sufficient knowledge about the characteristics of denture biofilms. Thus, the tolerance of DMP to existing antifungal drugs, its ability to evade components of the host’s immune system, and its resistance to the mechanical forces underlying the prosthesis make it a central subject of studies.

To summarize, Table 5 highlights the pertinent points that facilitate the differentiation between indications for PMMA, PA, and PEEK concerning DMP formation.
To limit the adhesion and accumulation of prosthetic microbial plaque, removable prostheses milled from a PMMA block best meet this requirement. Those made from PA are less efficient in terms of the colonization of microorganisms. As for PEEK, the long-term anti-adhesion properties seem to gradually diminish.Regarding the polishing and maintenance of removable prostheses, PMMA is also more efficient than PA and PEEK.Relined PMMA bases, both via thermosetting and machining, are easy to implement and effective.

This calls for the establishment of an effective strategic plan in the fight against persistent oral–prosthetic microbial infections that are likely to spread remotely via saliva or the bloodstream, such as DS.

## Figures and Tables

**Figure 1 polymers-16-00040-f001:**
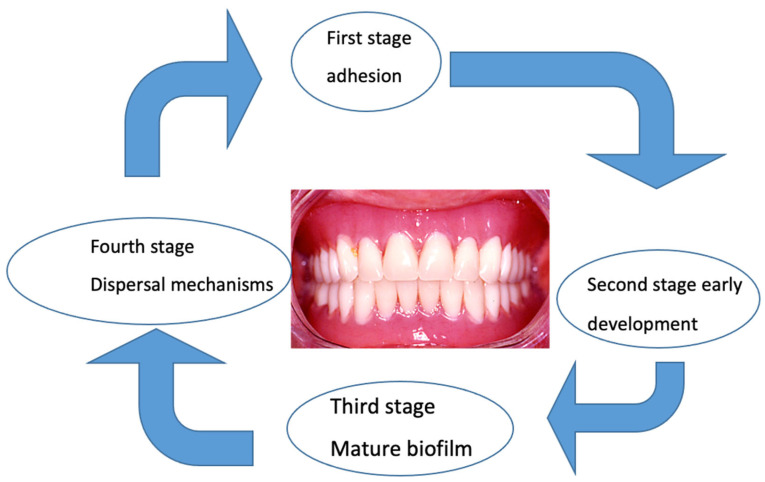
Biofilm envelops the denture in distinct stages. In the transition from the planktonic, free-floating state to the sessile state, attached microorganisms begin radically changing their gene and protein expression profiles.

**Table 1 polymers-16-00040-t001:** Chemical composition, surface roughness (Ra), and surface-free energy (SFE) of polymers (PMMA, PA, and PEEK) have the potential to affect bacterial adherence.

Polymers	Mean Surface Roughness (Ra) ± SD in µm Ra Threshold of 0.2 µm 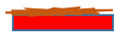	Surface-Free Energy (SFE) (N/m; mJ/m^2^) SFE Threshold of 40 mJ/ m^2^ 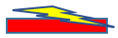	*C. albicans* Adherence 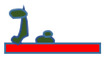	Bacterial Adherence 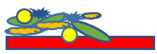
**PMMA**	**CAD/CAM PMMA:** Al-Dwairi ZN et al. 2019 [11]: 0.16 ± 0.03 μm (AvaDent PMMA billets; Global Dental Science, Scottsdale, AZ); Al-Dwairi ZN et al. 2019 [11]: 0.12 ± 0.02 μm (Tizian Blank PMMA; Schütz Dental, Rosbachvor der Höhe, Germany) Benli M et al. 2020 [12]: 0.19 ± 0.01 μm (Amann Girrbach AG, Koblach, Austria) Radford DR et al. 1998 [13]: 0.66 ± 0.34 μm (Trevalon Clear; Dentsply Ltd., De Trey Division, Weybridge, UK) Steinmassl et al. 2018 [14]: All CAD/CAM dentures had lower mean surface roughness values than conventional dentures. Steinmassl et al. 2018 [14]: −0.28 ± 0.16 µm AvaDent Digital Dentures (AD; Global Dental Science Europe BV, Tilburg, the Netherlands); Steinmassl et al. 2018 [14]: −0.44 ± 0.13 µm Baltic Denture System (BDS; Merz Dental GmbH, Lütjenburg, Germany); Steinmassl et al. 2018 [14]: −0.28 ± 0.01 µm Vita VIONIC (VV; Vita Zahnfabrik, Bad Säckingen, Germany); Steinmassl et al. 2018 [14]: −0.04 ± 0.01 µm Whole You Nexteeth (WN; Whole You Inc., San Jose, CA, USA) Steinmassl et al. 2018 [14]: −0.30 ± 0.10 µm Wieland Digital Dentures (WDD; Wieland Dental + Technik GmbH & Co. KG, Pforzheim, Germany/Ivoclar Vivadent AG, Schaan, Liechtenstein); Schubert et al. 2020 [15]: 0.07 ± 0.01 µm (Med 610 Stratasys, Eden Prairie, MN, USA); Schubert et al. 2020 [15]: 0.07 ± 0.01 µm (V-Print splint Voco, Cuxhaven, Germany); Schubert et al. 2020 [15]: 0.09 ± 0.01 µm FREEPRINT ortho 385 Detax, Ettlingen, G); Schubert et al. 2020 [15]: 0.06 ± 0.01 µm Dental LT Clear Formlabs, Somerville, MA, USA) Schubert et al. 2020 [15]: 0.06 ± 0.01 µm (M-PM crystal Merz Dental, Luetjenburg, Germany. 0.04 ± 0.01 Therapon Transpa Zirkonzahn, Gais, Italy) **Conventional heat-polymerized PMMA:** Al-Dwairi ZN et al. 2019 [11]: 0.22 ± 0.07 μm (Meliodent conventional PMMA, Heraeus Kulzer, Hanau Germany) Sultana N et al. 2023 [16]: 0.11 ± 0.04 μm (DPI Heat Cure; Dental Products of India, Mumbai, Maharashtra, India) Schubert et al. 2020 [15]: 0.04 ± 0.01 µm (Erkodur Erkodent, Pfalzgrafenweiler, Germany); Schubert et al. 2020 [15]: 0.05 ± 0.01 µm (PalaXpress ultra Kulzer, Hanau, Germany) Schubert et al. 2021 [15]: 0.04 ± 0.01 µm (Erkodur, Erkodent Pfalzgrafenweiler, Germany), Schubert et al. 2021 [15]: 0.05 ± 0.01 µm (PalaXpress ultra; Kulzer, Hanau, Germany) Steinmass et al. 2018 [14]: 0.55 ± 0.14 µm Candulor Aesthetic Red: Candulor AG, Glattpark, Germany) **Injection-molded technique:** Sultana N et al. 2023 [16]: 0.06 ± 0.02 µm (SR Ivocap High Impact; Ivoclar Vivadent AG, Schaan, Liechtenstein) Abuzar MA et al. 2010 [17]: 0.99 ± 0.12 μm before polishing (Vertex RS, Vertex-Dental BV, ZeiIthe Netherlands) which was reduced more than 20 times to 0.04 ± 0.007 μm after polishing.	**CAD/CAM PMMA:** Steinmassl et al. 2018 [14]: SFE mean values between 31.82 and 33.68 mJ/m^2^ for all the CAD/CAM dentures: AvaDent Digital Dentures (AD; Global Dental Science Europe BV, Tilburg, the Netherlands); Baltic Denture System (BDS; Merz Dental GmbH, Lütjenburg, Germany); Vita VIONIC (VV; Vita Zahnfabrik, Bad Säckingen, Germany); Whole You Nexteeth (WN; Whole You Inc., San Jose, USA); Wieland Digital Dentures (WDD; Wieland Dental + Technik GmbH & Co. KG, Pforzheim, Germany/Ivoclar Vivadent AG, Schaan, Liechtenstein) Steinmass et al. 2018 [14]: 66.62 ± 3.02 mJ/m^2^ WN with coating. Schubert et al. 2020 [15]: 68.44 ± 1.98 mN/m (V-Print splint Voco, Cuxhaven, Germany); Schubert et al. 2020 [15]: 69.81 ± 2.16 mN/m FREEPRINT ortho 385 Detax, Ettlingen, G); Schubert et al. 2020 [15]: 70.86 ± 0.31 mN/m Dental LT Clear Formlabs, Somerville, MA, USA); Schubert et al. 2020 [15]: 65.31 ± 0.88 mN/m (M-PM crystal Merz Dental, Luetjenburg, Germany. 63.66 ± 3.09 mN/m Therapon Transpa Zirkonzahn, Gais, Italy) **Conventional heat-polymerized PMMA:** Cabanillas B et al. 2021 [18]: −40.3 ± 0.3 N/m (PMMA; Vitacryl; A. Tarrillo Barba, Lima, Peru) Cabanillas B et al. 2021 [18]: −39.5 ± 0.3 N/m (PMMA; Triplex; Ivoclar Vivadent, Ellwangen, Germany) Steinmassl et al. 2018 [14): −33.00 ± 0.97 mJ/m^2^ conventional heat-polymerized resin (Candulor Aesthetic Red; Candulor AG, Glattpark, Germany) Schubert et al. 2021 [15]: −62.67 ± 3.43 mN/m Erkodur. Erkodent, Pfalzgrafenweiler, Germany) Schubert et al. 2021 [15]: −65.02 ± 2.41 mN/m PalaXpress ultra; Kulzer, Hanau, Germany) A decrease in SFE energy was observed for denture acrylic resins after storage in substances for the hygiene of dentures; calculated values of SFE (42.2–46.0 mJ/m^2^) showed the hydrophobic character of the surface and may increase bacterial adhesion (Rutterman 2011 [9]). Incubation conditions such as saline solution and substances for the hygiene of dentures had no significance impact on the SFE value. Liber-Kne’c, 2021 [10]. Gad MM et al. 2022 [19]: low SFE of denture base resin (hydrophobe)	**CAD/CAM PMMA:** Osman RB et al. 2023 [20] Schubert et al. 2018 [15]: in vitro CAD/CAM, 3D printing and milling increased the adherence of *C. albicans* compared to conventional manufacturing **Conventional heat-polymerized PMMA:** Cabanillas B et al. 2021 [18]: no difference between tow PMMA, the adhesion per cell/field of *C. albicans*, Vitacryl (A. Tarrillo Barba, Lima, Peru) presented 15.7 ± 1.1, N/m Triplex (Ivoclar Vivadent, Ellwangen, Germany) had 16.7 ± 2.3 N/m.	**CAD/CAM PMMA:** Steinmass et al. 2018 [14]: CAD/CAM dentures had smoother and more hydrophilic surfaces than conventional dentures; there was no difference in their free-surface energy except after coated dentures. Below the Ra threshold of 0.2 µm there was a slightl but insignificant correlation between Ra and microbial adhesion (*C. albicans* and *S. mutans*). **Conventional heat-polymerized PMMA** Moslehifard E et al. 2022 [21]: Injection vertex acrylic resin (Vertex Castaravia, Vertex Dental Zeist, the Netherlands) improved the decreased surface roughness of the denture base. Bacterial adherences decreased compared with the conventional method. (Vertex dental, Ist, the Netherlands)
**PA**	Abuzar MA et al. 2010 [17]: 1.11 ± 0.17 μm (Flexiplast, Bredent GmbH & Co KG, Senden, Germany) Sultana N et al. 2023 [16]: 0.19 ± 0.01 (Macro Flexi Dental Resin; Macro Dental World Pvt. Ltd., Jalandhar, Punjab, India)	Takabayashi 2010 [22]: hydrophilic nature Lucitone FRS.	Freitas-Fernandes 2014 [23]: highest *C. albicans* adherence on polyamide (Flexite MP/PMMA (Acron MC).	Sultana et al. 2023 [16]: The highest microbial adhesion was observed in injection-molded polyamide/PMMA.
**PEEK**	Benli M et al. 2020 [12]: 0.13 ± 0.01 µm (PEEK 100%; Amann Girrbach AG, Koblach, Austria) Batak B et al. 2021 [24]: Milled 100% PEEK were above 0.2 µm before and after polishing (Coprapeek; White Peaks Dental Systems GmbH & Co KG) Vulović S et al. 2022 [25]: Comparing PEEK (breCAM.BioHPP; Bredent group, Senden, Germany) and other CAD/CAM materials showed that samples of PEEK were slightly rougher than samples of PMMA. The reason for this could be related to the ceramic particles added to PEEK.	Hirasawa M et al. 2018 [26]: PEEK with lower SFE was hydrophobic and facilitated hydrophobic bacterial growth.	da Rocha LGDO et al. 2022 [27]: In vitro, hydrophobic *C albicans* facilite sessile yeast formation on the surface of PEEK/titanium alloy.	D’Ercole S et al. 2020 [28]: PEEK showed antiadhesive and antibacterial properties between 24 and 48 h against oral bacteria such as *Streptococcus oralis.* Ichikawa, T et al. 2019 [29]: Only one report showed clear plaque accumulation on the surface of claps PEEK after 2 years. Barkamo S et al. 2019 [30]: PEEK Ra of blasted surface > polished surface and facilited the adherence of bacteria, including *Streptococcus sanguinis*, *Streptococcus oralis*, and *Streptococcus gordonii.*

PMMA, polymethyl methacrylate; PEEK, polyether ether ketone; PA, polyamide; μm, micrometers. Higher SFE, improved wettability, and diminished CA reduce the adherence of *C. albicans*. A low value of SFE is sought to resist plaque in in vitro studies; conversely, high-energy surfaces collect more plaque and select specific bacteria. PMMA (injection-molded technique) showed better result than PA and PEEK for Ra, SFE, and bacterial adherence. A decrease in surface energy SFE was observed for denture acrylic resins after storage in denture hygiene substances, but calculated SFE values (42.2 to 46.0 mJ/m^2^) showed that they were hydrophobic.

**Table 2 polymers-16-00040-t002:** Different polishings of the polymers of the prosthetic base (mechanical polishing and/or chemical polishing in in vitro studies).

Polymers	Different Brands of Resin	Mechanical Polishing (MP) 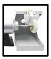	Chemical Manually Polishing (CP) 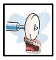	Findings and Results µm
**PMMA**	1—Heat-cured (HC) PMMA (Vertex RS Dentimex, the Netherlands); 2—prepolymerized block of CAD/CAM (Polident d.o.o. Volčja Draga 42, Sl-5293 Volčja Draga, Slovenia)	Polishing wheels, felt cones with pumice slurry, rubber polishers with RHPL and universal polishing paste (loose abrasives (aluminumoxide-Al_2_O_3_) in paste, Ivoclar Vivadent, Schaan, Liechtenstein), K50	Immersing them in a preheated jar at 75 ± 1 °C containing MMA monomer (Lang Dental Mfg. Co., Wheeling, IL, USA).	1—Heat-cured PMMA denture base material in both methods showed the highest mean Ra value (2.44 ± 0.07 and 2.72 ± 0.09 for MP and CP, respectively); 2—CAD/CAM denture base material showed the lowest mean values (1.08 ± 0.23 and 1.39 ± 0.31 for MP and CP, respectively) [31].
1—Heat cured (Probase Hot, Ivoclar Vivadent Inc., Schaan, Lichtenstein); 2—Probase Cold (cold-curing denture base, Ivoclar Vivadent Inc., Lichtenstein); 3—Palapress (Heraeus Kulzer, Hanau, Germany); 4—SR Ivocap (heat/pressure-curing (Ivoclar Vivadent Inc., Lichtenstein)	Polished with a mechanical milling system. The working tool speed was 5000 rpm as it progressed in the horizontal direction.	Manually polished.The polishing steps were the steps of ISO 20795 standard [32]	The Ra for the manually polished samples was globally significantly higher than for the mechanically polished samples [33].
1—heat-cured acrylic resin (Lucitone 199, Dentsply International, York, PA, USA); 2—auto-cured (AC) acrylic resin (Dentsply International Inc, York, PA, USA).	MP was performed with a felt-cone with pumice slurry and a wet felt-cone with caulk powder and water.	After finishing, the HC and AC specimens were immersed in MMA monomer heated approximately to 75 °C ± 1 °C for 10 secoIds.	The Ra in order of decreasing values were CP-HC: 1.41, CP-AC: 1.34, MP-AC: 0.73, and MP-HC: 0.63. MP was the most effective polishing technique for HC and AC resins [34].
1—HC: (Lucitone, Dentsply International Inc., York, PA, USA); 2—LC (light cured) acrylic resin (Eclipse, Dentsply Interna, Inc.). Both HC and LC prealably polishing was carried out with 360-grit sandpaper mounted on a lathe.	Performed with an automatic polishing machine (The Wirtz, Jean Wirtz, Dusseldorf W, Germany) for 2 min, under 50 rpm and 500 g of load with RHPL.	Performed by immersing the HC and LC specimens in Jet Seal Liquid (MMA at 75 ± 1 °C; Lang Dental Mfg. Co., USA).	RHPL, UPP, and K50 agents produced superior surface smoothness for all acrylic resin specimens and a mean Ra significantly below the threshold Ra of 0.2 µm. MP was the most effective polishing technique [35].
Three HC acrylic resin materials (1—DPI, 2—Meliodent, 3—Trevalon Hi) were grouped as Group A (unfinished), Group B (finished), Group C (polishing paste), Group D (polishing cake), and Group E (pumice and gold rouge).	Materials used: universal polishing paste (Ivoclar), polishing cake (Bego), pumice (micro-white, Asian chemicals), and gold rouge (Bego). Instruments used: felt cone, soft cloth wheels (which were prepared), polishing unit (Kavo), and a timer.	NR	Smoother surfaces were achieved with Trevalon HI, Meliodent, and DPI. The best results among the polishing materials came from the polishing paste, followed by the polishing cake, pumice, and gold rouge [36].
**CAD/CAM, HC, acrylic resin**CAD/CAM dentures (1–5) and conventional dentures (6). 1—standardized denture resin specimens: AvaDent Digital Dentures (AD; Global Dental Science Europe BV, Tilburg, Pays-Basque); 2—BalticDenture System (BDS; Merz Dental GmbH, Lütjenburg, Germany); 3—Vita VIONIC (VV; Vita Zahnfabrik, Bad Säckingen, Germany); 4—Whole You Nexteeth (WN; Whole You Inc., San Jose, CA, USA); 5—Wieland Digital Dentures (WDD; Wieland Dental + Technik GmbH & Co. KG); 6—Pforzheim, Germany/Ivoclar Vivadent AG, Schaan, Liechtenstein, with conventionally manufactured denture surfaces (control group).	NR	All dentures were manually finished. The mucosal surfaces were left unfinished and were examined.	All CAD/CAM dentures exhibited smoother and more hydrophilic surfaces than conventional dentures. Significant differences were found for AD, VV, WN, and WDD compared to the control group [14].
**CAD/CAM, HC, acrylic resin**1—CAD/CAM 3D-printed resin (3D) (CediTEC DB; VOCO GmbH, Germany); 2—CAD/CAM-milled resin (M) (V—Print dentbase; VOCO GmbH, Cuxhaven, Germany); 3—heat-polymerized resin (HP) (Probase^®^ Hot; Ivoclar Vivadent, Liechtenstein); 4—autopolymerized resin (AP) (Probase^®^ Cold; Ivoclar Vivadent, Liechtenstein); 5—injected molded resin (IM) (iFlexTM; tcs^®^, Signal Hill, CA, USA)	Mechanical technique with the Jota^®^ 1877 denture polish kit (Jota AG, Rüthi, Switzerland) protocol.	Manual technique with the Jota^®^ 1877 denture polish kit (Jota AG, Rüthi, Switzerland) protocol.	The resins submitted to manual polishing showed significantly lower mean surface roughness values than the control resin. CAD/CAM-milled acrylic resins demonstrated lower values of Ra compared to the conventional PMMA [37].
**PA**	1—HC PA (Vertex RS Dentimex, the Netherlands); 2-(Breflex, Bredent, Gmbh. Co.K.G. Senden, Germany); 3-CAD/CAM. (prepolymerized block acrylic resin denture base material (Polydent d.o.o. Volčja Draga 42, Sl-5293 Volčja Draga, Slovenia)	MP for HC, PA, and CAD/CAM specimens was performed using polishing wheels, felt cones with pumice slurry, rubber polishers with RHPL and universal polishing paste, and Abraso-Star K50 (K50) with light pressure for 15 s.	CP for HC, PA, and CAD/CAM specimens was performed by immersing them in a preheated jar at 75 ± 1 °C for 10 s.	PA surface roughness values: 1.77 ± 0.06 (MP) and 2.18 ± 0.10 (CP). PA contact angles: 67.90 ± 2.56 (MP) and 71.40 ± 2.50 (CP) (hydrophilicity). PA surface roughness values > CAD/CAM, PMMA values [31].
Polyamide (Valplast, Valplast International Corp., Long Beach, NY, USA)	NR	The polyamide was polished with Tripoli-Paste and Val-Mirror-Shine polishing paste (Weithas Corp., Lütjenburg, Germany).	Ra (PA 0.20 μm) did not change significantly after thermocycling or storage. Neither Ra nor the elasticity of PA was altered by artificial aging [38].
**PEEK**	High-purity (non-filler type) PEEK (JUVORA Dental Disc Invibio Biomaterial Solutions, Lancashire, UK)	NR	No additional polishing (NT), polishing using a Iber point (C), polishing using “silky shine” and a soft brush (S), polishing using “aqua blue paste” and a soft brush (A), protocol C followed by protocol S (CS), protocol C followed by protocol A (CA), protocol C followed by protocols S and A (CSA).	The PEEK polishing”chairside protocol produced clinically acceptable surface roughness, achieved using a brush and a mild polishing agent for more than 3 min [39].
PEEK (PEEK-IOF) BioHPP inorganic ceramicsand metal oxides	NR	Polished with 1000-grit SiC paper. A high-gloss finish was added using a 1 μm diamond paste applied with a cotton buff.	PEEK displayed the greatest change (increase) in contact angle values after air-polishing treatment. However, this effect could be prevented by veneering PEEK-IOF with DMA-nano components [40].
1—PEEK bioHPP (bredent Gmbh & Co. Press mode); 2—autopolymerizing denture PMMA (uniling PF 20, bredent Gmbh & Co. KG). All specimens were prepolished with a fine pumice stone (ERNST HINRICHIS Dental GmbH) and goat-hair brushes (bredent GmbH & Co. KG, Weissenhorner Str. 2, 89250 Senden, Germany).	Four laboratory polishing methods. 1—ABR: Abrasive polishing paste (bredent GmbH & Co. KG); 2—OPA: Opal L polishing paste (Renfert GmbH); 3—CER: Ceragum silicone polisher (bredent GmbH & Co. KG); 4—DIA: Diagen-Turbo, Ginder (bredent GmbH & Co. KG).	Three chairside methods: 1—SUP: Super-snap, polishing discs (Shofu dental GmbH); 2—PRI: prisma gloss, polishing paste (Dentsply De trey GmbH); 3—ENH: enhance, polishing system (Dentsply De trey GmbH).	Chairside polishing methods resulted in lower SR than laboratory-based methods. Both the SUP and PRI protocols led to PEEK surfaces with lower SR than ENH [41].

HC, heat-cured; LC, light-cured; MP, mechanical polishing; CP, chemical polishing; PMMA, polymethylmethacrylate resin; MMA, methyl-methacrylate monomer; DMA, dimethyl-methacrylate; PA, polyamide; PEEK, polyetheretherketone; SEM, scanning electron microscopy; AC, auto-cured denture base acrylic resins; RHPL, Resilit High-luster Polishing Liquid; UPP, universal polishing paste; K50, Abraso-Star K50; SR, surface roughness; AD, AvaDent; BDS, Baltic Denture System; VV, Vita VIONIC; WN, Whole You Nexteeth; WDD, Wieland digital dentures; DPI, dental promotion and innovation; CAD/CAM, computer-aided design/computer-aided manufacturing. CAD/CAM-milled acrylic resins had lower Ra values than heat-cured PMMA. Mechanical polishing of PMMA was superior to chemical polishing. Polishing the PAs made it possible to obtain a roughness close to 0.2 µm. Chairside polishing of PEEK also made it possible to obtain clinically acceptable values.

**Table 3 polymers-16-00040-t003:** Frequency and protocol for cleaning various polymer denture bases.

Polymer	Frequency of Use of Denture Cleansers (DC) against *Candida* spp. Adhesion 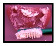	Denture Cleaning Treatment and Brushing Regimens (Night–Day) 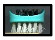	Ultrasonic Denture Hygiene 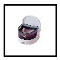	Tablet Composition May Directly Affect *C. albicans* Biofilm 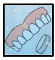	Other DisinfectantAgainst Inhibition of *C. albicans* (Oil, GSE, Ozone Neem) 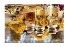	Immersion of Denture inChlorhexidine Digluconate (CHG)Sodium Hypochlorite (NaOCL) 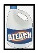
**PMMA**	The daily use of a DC overnight significantly reduced the total bacterial count [42,43,44,45,46].	The biofilm model on PMMA remained largely unaffected by brushing only [45]. Before overnight (8 h) storage conditions for limited colonization of *C. albicans* it is desirable to brush or use an alkaline peroxide-based tablet [47].	No statistically significant difference in total bacterial level between ultrasonic cleaning and brushing was found [44]. The adjunctive use of cetylpyridinium chloride with ultrasonic cleaning did not yield additional benefits [46].	Tow conventional heat-cured acrylic resins (1—QC-20 (Dentsply, Addlestone, UK), 2—Acron-hi^TM^ (Kemdent, Swindon, UK) and one polyamide (Deflex^TM^) were tested with the following solutions: 1—Polident 3 min™, 2—Corega™, and 3—Fittydent™. Polident 3 min^TM^ and Corega^TM^ tablets should be used for all denture resin types, whereas Fittydent^TM^ should only be proposed for those who use Deflex™ [48]. HC PMMA resin (Vertex-DentaIV., Zeist, the Netherlands) was subjected to a 4-week incubation with a daily change of 4 solutions (1—Clene^®^ (Bitec global group, Japan), 2—Polident^®^ (PTI Royston LLC, USA), 3—3% sodium bicarbonate (NaHCO3, Tianjin Lisheng Pharmaceutical Co. LTD, China), 4—phosphate-buffer Saline (PBS, Gi bco^®^ Invitrogen™, Cambridge, MA, USA)).Clene^®^, and Polident^®^ decreased fungal growth by approximately 98% and 100%, respectively [49].	Disc of 3D-printable resin (NextI Denture 3 D+, Soesterberg, The Netherlands) mixed with phytochemical-filled microcapsules and immersed in an effervescent tablet (Polident Quick, GSK Ireland) seemed to be a more effective inhibition of fungal cell growth compared with sterile tap water storage [50]. The treatment with 1% grapefruit seed extract (GSE) for 5 min almost eliminated the biofilm that formed on the resin [51]. Denture base PMMA (ACRON, GC, and Tokyo, Japan) immersed in ozone ultrafine bubble water (OUFBW) inhibited the early formation of *C. albicans* biofilms [52].	Specimens (acrylic resin Lucitone 550, Dentsply Ind. Com; Ltd.a., Petropolis, RJ, Brazil) were immersed for three cycles of 8 h in 2% CHG or 1% NaOCL. Residues of CHG were cytotoxic to gingival fibroblasts compare toIaOCL [53]. The in vitro effectiveness of five denture cleansers (Fittydent tablets, 2% CHG, 0.2% CHG, 0.5% and 1% NaOCL), was tested for microbial adhesion to the surface of base resins for conventional and CAD/CAM (milling and 3D printing) dentures: 1—conventional (Meliodent, Kulzer GmbH; Heraeus Kulzer Germany); 2—milling, Zintec CAD software (Wieland Digital Denture (Danbury, CT, USA)); and 3—3D-printed Denture I NextDent, Soesterberg, the Netherlands). The denture cleansers increased the roughness of all PMMAs. Concerning cleansers, the best result was obtained with 2% CHG and 0.5% and 1% NaOCL [54].
**PA**	After 20 days immersed in Corega Protefix, and Valclean, the highest surface roughness was observed in the Valplast polyamide resin. No difference was observed in PMMA resins (Paladent) [55].	Among the three commonly used DCs with different pH (Valclean—acidic, Clinsodent—alkaline, and Polident—neutral), Valclean showed statistically significant greater stain removal efficiency than Polident or Clinsodent [56].	Valplast was found to have a significantly lower gloss and a higher roughness than Paladon 65 before cleansing. After cleansing (control; Val-Clean, peroxide cleanser; Corega Extradent, peroxide cleanser), the gloss of both materials decreased and only the roughness of Paladon 65 increased [57,58].	The tested cleanser tablets were more effective for PMMA resin than for thermoplastic polyamide resin [59]. For patients who have polyamide-based prosthesis, the use of citric acid-based cleansers may be more recommended than sodium perborate [59].	Compared with Curaprox (eucalyptus oil, Curaprox, UK, Huntingdon, UK) the effervescent tablets (Corega, Protefix, Perlodent) significantly altered the surface hardness and roughness of the polyamide (Deflex-Nuxen SRL, Buenos Aires, Argentina) [60].	Thermo-injected polyamide dentureresin base colonized with *C. albicans* and disinfected with 0.12% chlorhexidine and Neem extract demonstrated the highest antimicrobial efficacy, with decreased surface roughness and no alteration in denture hardness [61].
**PEEK**	Daily DC recommended	Individual prophylaxis can be conducted with toothbrushes. For professional prophylaxis, air-abrasion devices using gentle powders are effective. Laboratory protocols should include gentle cleaning methods like ultrasonic bath [41].	PEEK prophylaxis in laboratory protocol includes gentle cleaning methods like ultrasonic bath [41].	Four dentures (1—SR Triplex Hot heat-polymerized PMMA (Ivoclar Vivadent AG., Schaan, Leichenstein); 2—SR Triplex cold auto-polymerized PMMA (Ivoclar Vivadent AG., Schaan, Leichenstein); 3—Deflex Injection molded polyamide, Nuxen —L, Buenos Aires, Argentina); 4—unfilled PEEK CAD/CAM Juvora Dental Disc (Juvora, London, UK) were tested.Three denture cleansers (DCs) after immersion for 120 days in a chemical solution applied to PEEK and other denture base materials (DBMs) on long-term water sorption and solubility were compared: Corega tablet (CT), Protefix tablet (PT) (Queisser Pharma, Flensburg, Germany), and 1% sodium hypochlorite (NaOCl) solution (SH) (Aklar Kimya, Ankara, Turkey), as well as a control (distilled water, DW). The PEEK group showed lower mean solubility values in DC than the other DBM groups. Auto-polymerized and injection-molded polyamide showed higher solubility [62].	Higher numbers of *Strep. oralis* and *C. albicans* on PEEK specimens confirmed the impact of the higher surface roughness and contact angle values on the microbial adhesion and described PEEK as less desirable than CoCr from a microbiological perspective [63].	PEEK seemed to be more stable against discolorations than other denture resin materials. Regarding the cleaning potential, individual prophylaxis can be conducted with toothbrushes. For professional prophylaxis, air-abrasion devices using gentle powders are effective. Laboratory protocols should include gentle cleaning methods like ultrasonic bath. Regarding the cleaning potential, individual prophylaxis can be conducted with toothbrushes. For professional prophylaxis, air-abrasion devices using gentle powders are effective [41,42].

PBS, phosphate-buffer saline; DC, denture cleanser; DCI, denture cleanliness index; CoCr, cobalt chromium; CFU, colony-forming units; GSE, Grapefruit seed extract; P, polident; OUFBW, ozone ultrafine bubble water. Daily frequency protocol (DC with brushing) for cleaning denture is recommended for the three polymers. The cleaning tablets tested proved effective for all PMMA and PEEK resins, whereas for the same result, thermoplastic polyamide resin required specific tablets. On the other hand, self-polymerized and injection-molded polyamide showed higher solubility than PMMA. Concerning PEEK, the cleaning tablets were effective with low solubility. Generally speaking, denture cleansers increased the roughness of all PMMAs. Concerning liquid cleansers, the best result was obtained with 2% CHG for CAD/CAM PMMA and 0.5% and 1% NaOCL for all PMMAs. Regarding thermo-injected polyamide-based resins colonized by *C. albicans*, disinfection with 0.12% chlorhexidine demonstrated the highest antimicrobial level.

**Table 4 polymers-16-00040-t004:** Comparison of different relining prosthetic base polymers (PMMA, polyamide, and PEEK).

Polymers	Relining Materials	Findings
**Unique relining materials**	Two heat-cured acrylic dentures (PMMA)—Lang (Lang Dental MFG Co., Wheeling, IL, USA) and VIx RS (Vertex Dental, Zeist, the Netherlands)—were prepared in the dental laboratory. Six silicone relining chairside self-cured materials were used: 1—Mucopren soft, (Kettenbach, es chenburg, Germany); 2—Mucosoft (Parkell, NY, USA); 3—Mollosil^®^ plus (Detax Ettlinghen, Germany); 4—Sofreliner Touch (Tokuyama, Tokyo, Japan),; 5—GC Reline™ Ultrasoft (GC Dental Products Co, Tokyo, Japan); 6—Silagum automix comfort (DMG, Hamburg, Germany). One self-curing (PEMA) chairside reline resin (Rebase II, Tokuyama, Tokyo, Japan) was used.	The contact angle increased for the materials in the following order: PMMA, PEMA, and silicone. The wettability of the denture relining except RebaseII and Mollosil^®^ plus was increased after water storage (24 h). The HC PMMA denture base showed the highest wettability. It can be suggested that heat-cured PMMA resin should provide superior denture retention and patient comfort than self-cured PEMA and silicone denture relining material [64].
**Conventional heat-polymerized PMMA** **PMMA and PMMA MMA pretreatment**	Soft liner type (silicone-based or PMMA-based)	The highest bond strength was observed in samples with silicone-based soft liners regardless of pretreatment. Silicone-based liners underwent adhesive failures, whereas PMMA-based liners underwent cohesives failures. In vitro exposure to *C. albicans* biofilms reduced the adhesion of denture liners to PMMA resin, suggesting that MMA pretreatment is recommended for relining procedures [65].
**CAD/CAM PMMA** **Dimethacrylate-based additively manufactured** **PMMA-based conventionally fabricated denture-base resins**	The tensile force applied to different materials was tested: 1—Heat-cured laboratory-side soft reliner; 2—Self-cured chairside soft reliner; 3—Self-cured chairside hard reliner.	The highest tensile bond strength was found between the conventional base and the self-cured chairside hard reliner (but no significant results were found with the laboratory-side reliner) [66].
**CAD/CAM PMMA** **3D-printed denture base by SLA method using DENTCA Denture Base II (DENTCA Inc., Torrance, CA, USA)**	Six surface treatment were applied to chairside relining materials with Tokuyama Rebase II Normal (PEMA) (Tokuyama Dental Corp, Tokyo, Japan): 1—no surface treatment (control); 2—Tokuyama Rebase II Normal adhesive (A); 3—Rocatec pre–sandblasting (Al_2_O_3_-110 µm)) (P); 4—Rocatec Pre + Tokuyama rebase II Normal adhesive (PA); 5—Rocatec Pre + ESPE silane (PS); 6—Rocatec system (Rocatec Pre + Rocatec Plus (Silica Al_2_O_3_-110 µm) + ESPE Sil (PPS).	The best adhesive and cohesive strength was obtained with the Rocatec system applied to a 3D-printed denture [67].
**CAD/CAM PMMA** **Conventional HC PMMA (ProBase Hot, Ivoclar Vivadent, Schaan, Liechtenstein)** **Milled Ivobase (Ivo-Base CAD for Zenotec, Wieland Dental, Pforzheim, Germany)** **Milled Ivotion (Ivotion A2/Pink V Denture Disc, Ivoclar Vivadent, Schaan, Liechtenstein)** **3D-printed group (NextDent DentuID+, NextDent B.V., Soesterberg, the Netherlands)**	Conventional relining PMMA resin (ProBase Cold, Ivoclar Vivadent, Schaan, Liechtenstein) monomer of the reliner (ProBase Cold Monomer, Ivoclar Vivadent, Schaan, Liechtenstein)	The shear bond strength of relined 3D-printed resins for a complete denture was lower than relined resins employed for CAD/CAM milling and conventional HC. When considering 3D-printing for CRDP fabrication, it is advisable to use it in clinical situations where frequent denture relining is not anticipated [68].
**POLYAMIDE** **Thermoplastic polyamide resin (Biotone; BT), injection mold (High Dental, Osaka Japan)** **Conventional heat-polymerized PMMA (Paladent 20; PAL20. Heraeus Kulzer, Hanau, Germany).** **Thermoplastic acrylic resin (Acrytone; ACT) (High Dental, Osaka, Japan)**	Tow chairside relining resins: Tokuyama Rebase II; TR II. PEMA (autopolymerizing polyethyl methacrylate) (Tokuyama Dental corp. Tokyo, Japan); Mild Rebaron LC, MRL, a light-activated PEMA (GC, Tokyo, Japan).	Among the three denture base resins, polyamide resin exhibited lower bond strength. However, no significant difference was observed for thermoplastic polyamide resin [69].
**POLYAMIDE/PMMA/CAD/CAM** **One polyamide (Vertex Thermosens, the Netherlands)** **One conventional PMMA (Meliodent HC, Kulzer, Hanau, Germany)** **Three PMMAs, CAD/CAM denture base material/subtractive** **(Ivoclar Vivadent, Schaan Liechtenstein)** **Polident pink disc basic, subtractive (Volcja draga, Slovenia)** **Anaxdent pink blank (U. Anaxdent GmbH, Germany)** **Two PMMAs CAD/CAM denture base material/additive (Freeprint Denture, Imprimo, Germany)** **Imprimo LC Denture, Iserlhon, Germany**	Two soft denture liners: Soft denture liner, acrylate-based, direct relining method (GC Europe, Leuven, Belgium); Reline II soft, silicone-based, direct relining method (GC Europe, Leuven, Belgium).	Relining polyamide denture base materials showed lower values of tensile bond strength with silicone-based soft liner than HC PMMA and subtractive denture base materials. The basic Polident pink CAD/CAM disc showed the highest tensile bond strength value in combination with the silicone-based soft liner [70].
**PEEK**	Surface treatment of PEEK begins with sulfuric acid etching, which promotes the highest bond strength, followed by air abrasion of the alumina particles. Then, the use of specific adhesives containing MMA, PETIA (pentaerythritol triacrylate), and dimethacrylates is recommended [71].	Sulfuric acid and alumina-particle air abrasion were the most effective surface treatments for promoting adhesion to PEEK. For clinical use, air abrasion with alumina particles can be considered the preferred solution [71]. In vitro PEEK presented a mixed type of failure involving adhesion and cohesion [72].

CRDPs, complete removable dental prosthesis; PMMA, polymethylmethacrylate resin; PA, polyamide; PEEK, polyetheretherketone; SR, surface roughness; PBS, phosphate-buffered saline; SBS, shear bond strength; RMGI, resin-modified glass ionomer cement. The best relining was obtained with conventional thermoset PMMA and a CAD/CAM-milling block. A specific system is necessary to obtain adhesive and cohesive strength with a relining 3D-printed denture base. PAs have low adhesion strength. PEEK is still under investigation, requires specific preparation, and exhibited a mixed type of failure involving adhesion and cohesion.

**Table 5 polymers-16-00040-t005:** Synthesis of the results concerning polymers (PMMA, PA, PEEK) for denture microbial plaque formation, polishing, relining, and hygiene.

Polymers	Microbial Adherence Treshold SFE: 40 mJ/m^2^	Polishing Treshold RA: 0.2 µm	Antimicrobial Nanoparticules	Relining	Cytotoxicity	Hygiene	General Condition
**PMMA**	Lower	Ra. Conventional injection-molded PMMA technique [16]. Sultana N et al. 2023 [16]: 0.06 ± 0.02 µm (SR Ivocap High Impact; Ivoclar Vivadent AG, Schaan, Liechtenstein) CAD/CAM acrylic resins demonstrated lower values of Ra compared to conventional PMMA [37].	The antifungal/antimicrobial effect of the material incorporated into the resin may have had a superior effect in preventing DS over simply coating the surface of the denture base [73].	The highest tensile bond strength was between the conventional base and the self-cured chairside hard reliner [66]. The shear bond strength of the relined 3D-printed resins for a complete denture was lower than the relined resins employed for CAD/CAM milling and conventional HC [68].	Lower level of cytotoxicity	PMMA is easy to maintain in the long term. CAD/CAM-milled prostheses are suggested in the presence of denture stomatitis due to reduced attachment of *Candida albican* [74].	Suitable for the majority of clinical indications, and small amounts of MMA for hypoallergenics patients
**Polyamide**	Higher	Clinicaly level > PMMA	Polyamide resin presented more viable cells of *Candida albicans*/PMMA [23].	Lower bond strengh	Toxicity profile	Difficult in the long term while respecting the manufacturer’s constraints	Temporary removable prosthesis, Parkinson’s disease, microstoma
**PEEK**	Intermediate	Chairside > laboratory method	Chitosan-based hybrid coatings on the PEEK surface contributed to the development of a biocompatible material (antibacterial, anti-inflammatory) [75].	Still under investigation	No evidence of cell damage caused by PEEK [76,77].	Efficacity decrease on the long term	Patients with low stress tolerance and sensitivity to metallic materials

Overall, PMMA presented advantages over PA and PEEK in most of the sections mentioned.

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
