# Peer review of "Different Polymers for the Base of Removable Dentures? Part II: A Narrative Review of the Dynamics of Microbial Plaque Formation on Dentures"

_polymers, 2023, doi:10.3390/polym16010040_

Round 1

Reviewer 1 Report

Comments and Suggestions for Authors

Good conclusion to previous manuscript. Should be ready for publication. 

Author Response

Comments and Suggestions for Authors

Good conclusion to previous manuscript. Should be ready for publication. Comments and

Response.  I thank the reviewer for his trust and approval.

Reviewer 2 Report

Comments and Suggestions for Authors

This narrative review described the denture-based materials for conventional, CAD/CAM milling, and 3D printing in terms of microbial plaque formation, especially Candida albicans, with focusing PMMA, PA, and PEEK. This paper is second part of the manuscript (Which polymer for the base of removable denture? Part I: A narrative review of mechanical and physical properties). The manuscript is of interesting for denture-based materials. However, the reviewer suggests that two review papers are merged into one manuscript. The contents written in these papers are not considered to be large enough to be published as two papers. A minor suggestion is to use Table or Figures. The present manuscript does not use figures and tables, which is inconvenient for a reader. Please utilize tables or figures as much as possible.

Comments on the Quality of English Language

Please checl the typos in the text.

Author Response

Reviewer 2;

Suggestions for Authors

This narrative review described the denture-based materials for conventional, CAD/CAM milling, and 3D printing in terms of microbial plaque formation, especially Candida albicans, with focusing PMMA, PA, and PEEK. This paper is second part of the manuscript (Which polymer for the base of removable denture? Part I: A narrative review of mechanical and physical properties). The manuscript is of interesting for denture-based materials. However, the reviewer suggests that two review papers are merged into one manuscript. The contents written in these papers are not considered to be large enough to be published as two papers. A minor suggestion is to use Table or Figures. The present manuscript does not use figures and tables, which is inconvenient for a reader. Please utilize tables or figures as much as possible.

 Response

Following the reviewer's suggestion, we produced a mini review on the subject (6931 words). So to respond to this request we have expanded the second article by including 7 tables (5 indexed in the text and two as supplementary materials). A final table allows you to summarize and finalize the results.

Reviewer 3 Report

Comments and Suggestions for Authors

Dear Authors,

Thank you for submitting this manuscript. I think the paper is quite interesting because it refers to a very important topic: which denture base material will be the best option for the fabrication of removable dentures, regarding denture microbial plaque formation. I would like to suggest some points to the Authors:

1. In the beginning, the abstract should include a short statement on the current research gap and the reasons to show why this study is unique and worthy of publication. The abstract should contain information as well about the literature sources, used search criteria, etc.

2. In the Introduction section, please add some information about the different types of denture base materials and their biocompatibility in the oral cavity. You can compare the different qualities of denture base polymers, regarding their manufacturing (heat-cured, CAD/CAM, etc.).

3. The Introduction section is very brief, you can add more information about the denture base materials, their properties, fabrication techniques, etc.

4. In the Introduction section, please add references from similar studies and reviews.

These references might be helpful to you:

Anadioti, E.; Musharbash, L.; Blatz, M.B.; Papavasiliou, G.; Kamposiora, P. 3D printed complete removable dental prostheses: A narrative review. BMC Oral Health 2020, 20, 343.

Dimitrova, M.; Corsalini, M.; Kazakova, R.; Vlahova, A.; Chuchulska, B.; Barile, G.; Capodiferro, S.; Kazakov, S. Comparison between Conventional PMMA and 3D Printed Resins for Denture Bases: A Narrative Review. J. Compos. Sci. 2022, 6, 87. https://doi.org/10.3390/jcs6030087

5. Since this is a narrative review, please add information about the sources of literature, including and excluding criteria, search databases, etc.

6. Where is your Results section? You have to report the significant findings from your study.

7. Where is your Discussion section? You have to discuss other authors' studies and compare their results with yours.

8. In the Conclusion section, please describe the significance of this study. The authors should summarize the significant findings in bullets for clarity in the Conclusion section.

I am sorry, but I am afraid this paper is of very poor quality, significant sections are missing. Therefore, I will have to reject this manuscript. Do not be discouraged, just make the necessary revisions, and resubmit it again.

Thank you in advance for all the corrections. Good luck!

Author Response

Reviewer 3

Comments and Suggestions for Authors

Dear Authors,

Thank you for submitting this manuscript. I think the paper is quite interesting because it refers to a very important topic: which denture base material will be the best option for the fabrication of removable dentures, regarding denture microbial plaque formation. I would like to suggest some points to the Authors:

  1. In the beginning, the abstract should include a short statement on the current research gap and the reasons to show why this study is unique and worthy of publication. The abstract should contain information as well about the literature sources, used search criteria, etc.

Response

To meet the reviewer's request, the summary has been embellished with a brief statement on current research gaps. An explanation is provided on the interest motivating the merit of publishing this synthesis of the literature on a current topic. The summary is also supplemented by information on the sources of the literature and the search criteria used.

  1. In the Introduction section, please add some information about the different types of denture base materials and their biocompatibility in the oral cavity. You can compare the different qualities of denture base polymers, regarding their manufacturing (heat-cured, CAD/CAM, etc.).

Response. We added comments about denture base materials, fabrication techniques

  1. The Introduction section is very brief, you can add more information about the denture base materials, their properties, fabrication techniques, etc.
  2. In the Introduction section, please add references from similar studies and reviews.

Response

These references might be helpful to you:

Anadioti E, Musharbash L, Blatz MB, Papavasiliou G, Kamposiora P. 3D printed complete removable dental prostheses: a narrative review. BMC Oral Health. 2020 Nov 27;20(1):343. doi: 10.1186/s12903-020-01328-8. PMID: 33246466; PMCID: PMC7694312.

 Dimitrova, M.; Corsalini, M.; Kazakova, R.; Vlahova, A.; Chuchulska, B.; Barile, G.; Capodiferro, S.; Kazakov, S. Comparison between Conventional PMMA and 3D Printed Resins for Denture Bases: A Narrative Review. J. Compos. Sci. 2022, 6, 87. [CrossRef]

Response. Both references have been added in the introduction.

  1. Since this is a narrative review, please add information about the sources of literature, including and excluding criteria, search databases, etc.

Response .We have added a paragraph on materials and methods.

  1. Where is your Results section? You have to report the significant findings from your study.

Response . We have added a paragraph on the results with 5 tables and 2 supplements.

  1. Where is your Discussion section? You have to discuss other authors' studies and compare their results with yours. Where is your Discussion section? You have to discuss other authors' studies and compare their results with yours.

Response. After the paragraph on the results we introduced a discussion which reviews the different tables.

  1. In the Conclusion section, please describe the significance of this study. The authors should summarize the significant findings in bullets for clarity in the Conclusion section.

Response. In the conclusion, the reference to Table V allows us to summarize the results.

I am sorry, but I am afraid this paper is of very poor quality, significant sections are missing. Therefore, I will have to reject this manuscript. Do not be discouraged, just make the necessary revisions, and resubmit it again.

Thank you in advance for all the corrections. Good luck!

Response. We believe we have followed your advice and comments and hope to have improved the article.

Round 2

Reviewer 2 Report

Comments and Suggestions for Authors

The manuscript has been improved.

Comments on the Quality of English Language

Please check the manuscript again.

Reviewer 3 Report

Comments and Suggestions for Authors

The authors have made extensive revisions and have improved significant the manuscript. The comprehensive exploration of polymer options for removable denture bases in Part II offers valuable insights into the dynamics of denture microbial plaque formation. The narrative review not only delves into the various polymers but also sheds light on their impact on oral health. This thorough examination equips both professionals and patients with knowledge crucial for making informed decisions, contributing to improved denture longevity and overall oral well-being. Overall I support the publication of this article. 

Thank you for the improvements.

Best regards